# Behavior-Based Control for an Aerial Robotic Swarm in Surveillance Missions

**DOI:** 10.3390/s19204584

**Published:** 2019-10-21

**Authors:** Pablo Garcia-Aunon, Jaime del Cerro, Antonio Barrientos

**Affiliations:** Centre for Automation and Robotics (CAR), Technical University of Madrid (UPM), 28006 Madrid, Spain; j.cerro@upm.es (J.d.C.); antonio.barrientos@upm.es (A.B.)

**Keywords:** robotic swarm, Multi-UAV, surveillance, behavior-based control, experimental results

## Abstract

Aerial robotic swarms have shown benefits for performing search and surveillance missions in open spaces in the past. Among other properties, these systems are robust, scalable and adaptable to different scenarios. In this work, we propose a behavior-based algorithm to carry out a surveillance task in a rectangular area with a flexible number of quadcopters, flying at different speeds. Once the efficiency of the algorithm is quantitatively analyzed, the robustness of the system is demonstrated with 3 different tests: loss of broadcast messages, positioning errors, and failure of half of the agents during the mission. Experiments are carried out in an indoor arena with micro quadcopters to support simulation results. Finally, a case study is proposed to show a realistic implementation in the test bed.

## 1. Introduction

### 1.1. Robotic Swarms

Multi-agent, or more specifically, multi-robot systems present obvious advantages over their single-agent counterpart. Generally speaking, the benefits of using these systems rely on the fact that sub-dividing a task usually leads to *(i)* accomplishing it faster by parallelizing it and/or using specialized agents; *(ii)* improving reliability and robustness due to redundancy; *(iii)* simpler and probably less expensive agents; *(iv)* accomplishing some tasks that a single-agent system would not be able to perform. However, these systems have as well fundamental drawbacks that must be properly addressed. In the case of robots that physically interact between them, sharing a common space, the main concern is how to properly coordinate them [1].

In the group of multi-robot systems, the robotic swarms have their own specific characteristics [2]: autonomous robots (sensor networks should not be considered in this group), large number of robots, few homogeneous groups, relatively incapable individually, and with local sensing and communication capabilities. Inspired by some animal behaviors and simple organisms, these algorithms perform complex task taking simple individual decisions, and sharing information and coordinating with other robots in a very simple and local way. As result, robotic swarms show very high robustness, flexibility and scalability, compared with other types of multi-robot systems. The fact that an intelligence must arise as a result of simple and local interactions is usually problematic, since designing the control of each robot individually turns out to be difficult.

Historically, most of the past works with robotic swarms have addressed different problems, among which we can name [3]: aggregation (gathering around a common place), flocking (jointly moving preserving distance between the robots), foraging (collecting items and bringing them to a common base), object clustering (group objects scattered in the scenario) and path formation (building a path with the robots between two given points). In a large majority of these works, only ground robots have been considered. A smaller group of past works have used Unmanned Aerial Vehicles (UAVs), either fixed- and rotary-wing aircraft.

### 1.2. Aerial Swarms

Because of safety reasons and movement capabilities, aircraft are usually aimed at working in outdoor environments. Additionally, because of their inherent nature, they can overcome space limitations that their ground counterparts can not (for example, flying over a terrain with multiple obstacles). Hence, the space assigned to each robot individually is usually much bigger in the case of aerial robots, and the distance between them is therefore also bigger: they are more spread in the scenario. Local interactions between UAVs are then difficult and more elaborated communication capabilities are usually considered.

Considering this argument, the tasks addressed with aerial swarms are usually very different from the ones solved with ground robots [4]. Flocking, or flying in formation, has been deeply investigated in many works, such us in [5] and [6], where the position of each agent is known and shared with the others.

Search and tracking of static or moving objectives have been also studied in the past. For instance, in [7], a group a UAVs moves coordinately to search for a source of pollution in an open space while each member keeps a safety distance with the surrounding agents. In [8] a swarm of aerial robots search for targets with random walks and preserving flying bearings; when a target is found, a virtual pheromone is deposited, which attracts other agents to it.

Surveillance, task in which this work is focused, has been also studied in the past with aerial swarms. Previous works addressing this problem will be analyzed in the following section.

### 1.3. Surveillance with Aerial Swarms

When addressing the surveillance of a continuous area with a group of robots (either considering swarms or multi-robot systems), we can distinguish three different problems with their own requirements: coverage surveillance, persistent monitoring (a.k.a. persistent surveillance) and surveillance.

The first of the three, coverage surveillance, comprises the algorithms that try to find optimal positions for the robots so that the area of interest is covered as good as possible. Often the same algorithms can be simply defined as robot deployment. In some cases, the importance of specific areas or the validity of the trajectories to reach those points is taken into account. For instance, in [9], an evolutionary-based optimization algorithm finds optimal positions and generates feasible trajectories to reach them, considering scenario constraints and limitations of the UAVs. Although many of these works require a centralized control architecture, others (such as in [10]) propose approaches that are fully distributed.

In case the area of interest is much bigger than the area that can be instantaneously observed by the group of agents, it is needed that they somehow move along the area to gather the information. Moreover, that information must be updated frequently, so that the agent must revisit the places repeatedly. In those cases, the problem is referred as surveillance. If there are specific hard requirements regarding maximum or mean age of the information, the problem is referred as persistent surveillance, or persistent monitoring. As it will be shown in the following section, the problem here addressed belong to this last group, since the mean age of the information is key to assess the efficiency of the algorithm.

In [11], the surveillance is accomplished by dividing the irregular area in portions and assigning them to each aerial agent in a distributed way, using one to one communications. The proposed approach is simple, scalable, easy to configure, it shows certain robustness against agents failures, and has low communication needs. However, it also requires spacial synchronization between the agents, their movements are predictable (which may be undesirable in some scenarios), and its adaptation to changes in the scenario is expected to be slow.

In other works, such as in [12], the agents plan their own trajectories to minimize a cost function, which takes into account the uncertainty of the information (through the time elapsed since the information was gathered) and the risk of flying at low altitudes. Moreover, altitude is selected to maintain a good data quality with low risk of colliding against the ground. The agents are therefore naturally attracted to non visited areas, being necessary sharing the uncertainty map between the agents. The algorithm shows flexibility in case any agent leaves or joins the group because the shared uncertainty map is continuously updated.

The maximum age of the information associated with the cells of a 2D grid is used in [13] to lead the UAVs. In a greedy fashion, each agent select which cell to visit next so that the ones with higher age are chosen first. The algorithm includes an arbitration procedure so that two agents do not travel to the same cell at the same time, and a policy in case the remaining fuel in a UAV gets low; in that situation, the UAV will try to stay close to a base-station. The work is primary aimed at reducing the maximum age present the area and it also includes real tests carried out in a indoor arena with up to 4 quadcopters.

In [14], the aerial robots behave like a flock and the group is guided to more interesting areas (that is, areas with older information) making use of digital pheromones. Additionally, the flock has the capability of avoid obstacles. Similarly in [15], a field of pheromones are deposited in the surveillance area, which attracts the agents to the areas with higher information age.

### 1.4. Contribution of This Work and General Structure

In this work, we present a behavior-based algorithm to guide a group of $N_{a}$ agents during a surveillance mission in a rectangular area. This algorithm has been designed based on a previous algorithm that was designed for searching instead of surveillance. In [16] the former algorithm was presented and compared with heuristic strategies for searching considering different number of agents, scenario size and aspect ratio, speeds, and size of the sensor. How the algorithm can be configured for those variations was studied in detail in [17]. Finally, in [18] and [19] the algorithm was modified to monitor the traffic in a simulated city within the project SwarmCity (http://www.swarmcityproject.com/). The main differences between the algorithm designed for searching and the here-used one are two. First, the pheromones-based behavior (see Section 3.1) has been simplified, removing two of the pheromone layers. Secondly, the optimization process uses a different fitness function, which tries to minimize the mean age of the information of the surveillance area.

The contributions of this work are the following:Adaptation of the algorithm for pure surveillance with minor changes.Robustness analysis against communication message losses, positioning errors, and failures of the agents.Experiments in an indoor arena to support simulated data and demonstrate implementation.A realistic use of the algorithm showed by a case study.

The proposed method is fully distributed and has low communication requirements between the agents. It is scalable and it can adapt to different scenarios, including areas with higher priority and obstacles.

This document is organized as follows: in Section 2 the surveillance problem is formulated, specifying the scenario and the agent model (a quadcopter), and the procedure to evaluate the efficiency is shown. The algorithm itself is described in Section 3, briefly indicating how it was optimized. Our approach is then tested in simulation and the results shown in Section 4, whereas in Section 5 the robustness of the algorithm is tested and analyzed in simulation. In Section 6 the test bed where the experiments were carried out is detailed, as well as the results of the experiments. Finally, in Section 7, a case study is presented, where real quadcopters try to detect intruders in an area of interest.

## 2. Analysis of the Proposed Task

The task at hand is persistently search in a rectangular area with fixed dimensions. A number of $N_{a}$ quadcopters can fly during a certain period of time at a nominal speed $V_{n}$. At each moment, each quadcopter can observe with a downward camera a circular area under it with a radius of $R_{f}$ meters, which is usually known as sensor footprint.

Keeping in mind that the algorithm will be finally tested with real drones in an indoor arena with certain dimensions, the surveillance area and quadcopter model characteristics will be accordingly chosen. In a real world application, quadcopters, surveillance area and sensor would be different. However, the results here shown can be extrapolated to a real world scenario.

### 2.1. Surveillance Area

The area under surveillance is a rectangular and planar surface with dimensions $L_{x}=6.4\times L_{y}=4.4$ m, with a total area of $A_{s}=24.4$ m${}^{2}$. To account which parts of the surface have been observed and when, the area has been subdivided into squared discretization cells with a length of 0.02 m. At the same time and because the surveillance algorithm requires it, the area has been additionally subdivided into a so-called search cells. Those squares have a length ΔL such as they can be exactly inscribed inside the sensor footprint, that is:(1)$$\Delta L=\sqrt{2}\cdot R_{f}$$

The agents move continuously from the center of one search cell to the center of other. Each time an agent reaches a cell, it decides which of the 8 surrounding cells will visit next. The agents fly at a constant altitude of h= 1.5 m. In Figure 1, a schematic view of the surveillance area and its main parameters have been represented.

### 2.2. Model of the Qutadcopter

Quadcopters are usually modeled in detail through their dynamic equations, considering the aerodynamic forces of the rotors and their inertia a mass properties. However, in this work, only the linear velocity response of the quadcopters used in the experiments is needed.

The model proposed in this work is a first-order time-delay system identified using a data set generated in the arena with the real quadcopters. As it will be further detailed in the experiment section, the quadcopters used are the Parrot Mambo Minidrones (see Section 6). The considered transfer functions (first-order time-delay) of the velocity in the body frame (*b*) are:
(2a)$$\begin{array}{c} \frac{V_{x}^{b}}{u_{x}}=\frac{b_{x}\cdot e^{-T_{d}\cdot s}}{s+c_{x}} \end{array}$$
(2b)$$\begin{array}{c} \frac{V_{y}^{b}}{u_{y}}=\frac{b_{y}\cdot e^{-T_{d}\cdot s}}{s+c_{y}} \end{array}$$
where $u_{x}$ and $u_{y}$ are the system inputs, whose values are calculated as follows:
(3a)$$\begin{array}{c} u_{x}=-\tan\left(\theta \right)=\tan\left(\alpha_{max}\cdot cmd_{x}\right) \end{array}$$
(3b)$$\begin{array}{c} u_{y}=\tan\left(\phi \right)=\tan\left(\alpha_{max}\cdot cmd_{y}\right) \end{array}$$
being θ and ϕ the pitch and roll angle, $\alpha_{max}=20^{\circ }$ the maximum allowed attitude angle, and $cmd_{x}$ and $cmd_{y}$ the commands in each body direction. Both commands have a range of [−1, 1], and when multiplied by $\alpha_{max}$ result in the desired pitch and roll angle. Inputs $u_{x}$ and $u_{y}$ are then the projection factor of the propellers force into the *x* and *y* directions. The commands are calculated applying a PID controller over the desired velocity. In Table 1 the parameters of the model have been presented. Each time a new simulation starts, the values on the table are withdrawn from a Gaussian distribution considering the %95 error intervals shown in the table.

Besides the velocity time response, it is necessary to define the nominal speed with which the quadcopters will fly, the sensor footprint radius and a safety region around the quadcopter to avoid collisions between them. As already mentioned, the drones used are the Parrot Mambo Minidrones, with an external dimension of 0.18 × 0.18 m. A safety radius of $R_{safety}=0.30$ m will be therefore set. This radius indicates the circumference inside which any other quadcopter shall not enter to preserve safety.

The next step would be selecting an appropriate size of the sensor footprint. Given that the size of the search cells (see Figure 1) is closely related to the size of the sensor footprint through Equation (Equation 1), the number of cells inside the area will be consequently related with that parameter as well. The higher the number of cells is, the higher the number of possible movements will be, and therefore the complexity of the scenario. Therefore, the study of the algorithm becomes more interesting with an increasing number of search cells, that is, with a low sensor footprint. On the other hand, there is a lower bound of $R_{f}$: lower values will provoke that two agents could not reach two adjacent cells at the same time because they would collide. Hence, the sensor footprint radius will be selected as small as possible but leaving some extra space:
(4a)$$\begin{array}{c} \Delta L>R_{safety}=0.30\;m \end{array}$$
(4b)$$\begin{array}{c} R_{f}=\frac{\Delta L}{\sqrt{2}}>\frac{R_{safety}}{\sqrt{2}}=\frac{0.30}{\sqrt{2}}=0.21\;m\to R_{f}=0.30\;m \end{array}$$
which leads to ΔL=0.43 m. Selecting a lower sensor footprint may lead to undesired interferences between agents flying to adjacent cells. Bigger sensor footprints are possible, but would reduce the number of cells (having fixed the size of the area) and then the complexity of the scenario.

The last parameter to be set is the nominal flying speed, $V_{n}$. Having tested the real quadcopters in the test bed, it was verified that speeds lower than 0.10 m/s were difficult to be correctly controlled. On the other hand, considering that the quadcopters have to fly between the centers of the search cells (with a size of 0.43 × 0.43 m), speeds higher than 0.15 m/s provoke that the agents could hardly keep the track between the cell centers. Consequently, in this work, two nominal speeds will be considered, $V_{n}=$ 0.10 m/s and $V_{n}$ = 0.15 m/s.

### 2.3. Measuring the Performance

Surveillance implies periodically visiting and observing the points inside the area of interest. Assuming that we can not observe the complete space at the same time, if we want to gather information from all the points, the sensors (in this case, the quadcopters) must move throughout the area. This implies that the information we have at each time about the complete area will be collected in different moments: in a certain instant, some information will be completely updated, whereas other will have a certain “*age*”, or elapsed time since that information was obtained.

When measuring the performance of a system for a surveillance task, it is often used the age of the information. The lower the age is, the better the surveillance system is because the information is more updated. In this work, the mean age of all the discretization cells is considered as a measurement of the effectiveness of the system, and it is simply calculated as:(5)$$\bar{a}\left(t\right)=\frac{1}{N_{d}}\sum_{i=1}^{N_{d}}a_{i}\left(t\right)$$
where $a_{i}\left(t\right)$ is the age of the information associated to the discretization cell *i* at time *t*, $N_{d}$ is the total number of discretization cells, and $\bar{a}\left(t\right)$ is the mean age of the area at time *t*. Assuming that $\bar{a}$ changes over time, we can further evaluate the age of the information during a complete mission:(6)$$\bar{A}\left(t\right)=\frac{1}{t_{m}}\int_{0}^{t_{m}}\bar{a}\left(t\right)dt$$
where $t_{m}$ is the total duration of the mission.

With Equations (Equation 5) and (Equation 6) we can measure how old the information is on average during the mission, but it would not indicate objectively how efficient our system is. Let us suppose that the system in composed by a group of $N_{a}$ agents, with a nominal speed $V_{n}$ and a sensor footprint $R_{f}$. Using a certain algorithm, whose efficiency is considered to be independent from the number of agents used, we obtain a certain age during the mission, ${\bar{A}}_{r}$. If we duplicated the number of agents, we can assume that the mean age would be reduced to a half, ${\bar{A}}_{r}/2$. Something similar could happen if the nominal speed or the sensor footprint were duplicated, or if the surveillance area were reduced to a half. Therefore, the mean age can not be used directly as a measurement of how efficient an algorithm or a method is because it depends on the resources assigned to the surveillance system (number of quadcopters, for instance) and on the area of interest.

To objectively measure the efficiency, an ideal mean age must be found to normalize the mean age obtained. In previous works [16] it was shown that the minimum time to observe the area only once was:(7)$$t_{min}=\frac{A_{s}-A_{i}}{2R_{f}V_{n}N_{a}}$$
being $t_{min}$ the minimum time needed to observed only once an area of $A_{s}$ [m${}^{2}$] with $N_{a}$ quadcopters flying at $V_{n}$ [m/s] with a sensor footprint radius of $R_{f}$ [m]. $A_{i}$ is the observed area at t= 0, which usually is $A_{i}=N_{a}\pi R_{f}^{2}$. When the area has been completely observed once ideally, the zones that were observed in first place will have an age of $t_{min}$ as per Equation (Equation 7), whereas the last ones will have an age of 0 seconds. Therefore, assuming a constant flying speed, the mean age of all the cells in the area will be:(8)$${\bar{a}}_{min}=t_{min}/2$$
where ${\bar{a}}_{min}$ is the minimum mean age that can be achieved with a given system for a given area size. If the area keeps being observed, we can assume that the minimum mean age during the complete surveillance mission will be:(9)$${\bar{A}}_{min}={\bar{a}}_{min}=t_{min}/2=\frac{A_{s}-A_{i}}{4R_{f}V_{n}N_{a}}$$

With the above result we can now define an objective way of measuring the efficiency of an algorithm in the established scenario. If an algorithm achieves a mean age ${\bar{A}}_{r}$ during a certain period of surveillance time, the efficiency will be then computed as:(10)$$E=\frac{{\bar{A}}_{min}}{{\bar{A}}_{r}}$$
which will be always lower or equal than 1.

The only issue left in the above formulation is how to initialize the age of the cells. If at the beginning of the task the age of the cells is set to 0, the initial efficiency at t=0 would be infinite. On the other hand, setting it to a very large quantity would lead to an initial efficiency close to 0. Since the mission efficiency is calculated as an average over time, both solutions would have a non-realistic impact on it. Actually, if the initial age is set to infinity, the efficiency would remain equal to 0 until all the cells are observed at least once, which would have an important impact on the final efficiency. Knowing that the final efficiency (based on experience) of the configured algorithm is close to 0.6, the age of the cells would be initialized to:(11)$$a\left(t=0\right)=\frac{1}{0.6}\cdot {\bar{a}}_{min}=\frac{1}{0.6}\cdot \frac{A_{s}-A_{i}}{4R_{f}V_{n}N_{a}}$$
This way, the initial efficiency is forced to be equal to 0.6. Selecting the initial age in this way is a commitment decision so that it is expected to have a low impact on the final efficiency of the algorithm over the complete mission.

## 3. Description of the Algorithm

In this section the control algorithm, which guides the movements of the agents, will be presented. Originally, the algorithm was designed for searching tasks [16], and it was shown how the algorithm could be configured to adapt to different scenarios [17]. Moreover, the algorithm was modified to carry out surveillance tasks in other works [18,19]. Given that the version of the algorithm used in this work does not largely differ from past implementations, only a general summary will be here discussed.

The algorithm is designed to work distributively. Each agent broadcasts certain information about its state and receives the similar messages from the other agents. With the received information, the algorithm is executed on board and when the agent reaches the center of a search cells, it decides which cell to visit next from the 8 surrounding ones. It has 2 main and differentiated parts: a behavior based control (the core of the algorithm) and a low level control, which includes a collision avoidance algorithm. In Figure 2, a complete overview of the control has been shown.

### 3.1. Behavior-Based Control

The behavior-based control algorithm is composed by 6 behaviors. Each time an agent reaches a cell, it has to decide which cell to visit next. Each behavior evaluates in that moment the 8 surrounding cells and assigns values to them. All this values are summed up applying weights by a final decision module, which finally selects the cell with higher value.


*Pheromones*


The pheromones behavior is the main one and it is in charge of leading the agents to areas with higher age. Similarly to the equivalent behavior implemented in past works, the discretization cells generate pheromones as time goes by. The concentration of pheromones is spread out in the scenario following the diffusion equation, with a diffusion coefficient to be set. As the agents observe the area, they remove a certain percentage of pheromones concentration. The main difference between the implementation in this work compared with the past ones, is that there is only one layer of pheromones. In Figure 2, the pheromone behavior has been labeled with 1. The output of the behavior is related to the concentration of pheromones inside each search cell.


*Coincident cells*


It may happen that two close-enough agents decide to go to the same cell, in which case a collision might occur. To avoid this situation, a coincident cell behavior has been implemented. This behavior penalizes cells that have been previously selected by other agents as next cells. In Figure 2, the coincident cells behavior has been labeled with 2.


*Energy saving*


The energy saving behavior has two main purposes. On the one hand, it tries to minimize the energy spent by trying not to change the flying direction. On the other hand, given the dynamics of the quadcopters, not changing the current direction helps in following the established tracks between the center of the cells. Therefore, this behavior penalizes movements that implies higher flying direction changes. In Figure 2, the energy saving behavior has been labeled with 3.


*Diagonal movement*


The surveillance space has been divided in cells that are inscribed inside the sensor footprint. Because of this, when an agent moves in the *x* of *y* direction, there is an overlap of a part of the sensor footprints with the adjacent cells. However, if the complete area were swept following diagonal movements, there would be no overlap, which would be more efficient. Due to this fact, this behavior reinforces diagonal movements. In Figure 2, the diagonal movement behavior has been labeled with 4.


*Keep distance*


Based on a virtual potential field, this behavior controls that the distance between the agents is appropriate. Inspired by the intermolecular attractive-repulsive forces, it creates a virtual force whose sign and magnitude depends on the distance to each agents, punctuating the surrounding cells accordingly. In Figure 2, the keep distance behavior has been labeled with 5.


*Keep velocity*


Inspired by how the bird flocks behave, this behavior aims to keep the velocity of the agents similar. The idea is that if some agents are flying in a given direction (presumably because the are traveling to areas with higher ages), surrounding agents are enforced to follow that direction. In Figure 2, the keep velocity behavior has been labeled with 6.


*Final decision*


Once each surrounding cell has been rated by each behavior, the final decision module applies a weight to each value, and sums them up:(12)$$I_{f_{g}}=I_{P_{g}}+C_{C_{g}}+\alpha_{E}C_{E_{g}}+\alpha_{DM}I_{DM_{g}}+\alpha_{D}I_{D_{g}}+\alpha_{V}I_{V_{g}}$$
where the α coefficients are the weights associated with each behavior, and the subindex *g* indicates the cell, g=1,…,8. The module then selects the cell with higher score and it is passed to the low level control and collision avoidance module.

### 3.2. Low Level Control and Collision Avoidance

Once the goal cell has been selected, the low level control generates a velocity command in global axis to fly from the current position to it. Therefore, this module requires the current position of the agent in the global axis *O*. The velocity is then transformed into the body frame and a PID controller is then applied to finally generate the commands.

Although the coincident cell behavior (see Figure 2, labeled with 2) may avoid some collisions between the quadcopters, in some other situations an additional mechanism must be implemented to avoid an impact. In Figure 3, two examples of these type of situations have been represented.

To resolve such situations and looking for fast reactivity and simplicity, another approach has been implemented, which is based on two mechanisms: a virtual force and an exchange cell protocol.


*Virtual force*


As extensively done in the past [20], the first mechanism to avoid collisions between agents is a virtual force. It is meant to keep a safety distance between the agents while they head to their respective goal cells, trying to interfere as less as possible and not to provoke unwanted deadlocks.

If an agent *A* detects a conflict with agent *B*, it first calculates its relative velocity with the other agent:(13)$$V_{A}^{B}=V_{A}^{0}-V_{B}^{0}$$
where subscript refers to the agent and the superscript to the reference frame. The reference frame attached to *B* has been defined centered on it and with its axes parallel to the inertial frame, named *O*. Secondly, a virtual force $F_{A}^{B}$ is created, which has two normal components:
(14a)$$\begin{array}{c} F_{A}^{B}=V_{n}\cdot \alpha_{c}\cdot {\left(1-\frac{|r_{AB}|}{\rho_{0}}\right)}^{\mu_{c}} \end{array}$$
(14b)$$\begin{array}{c} F_{A}^{B}=-F_{A}^{B}\cdot \left(n_{A}^{B}+t_{A}^{B}\right) \end{array}$$
where $V_{n}$ is the nominal speed of the agents, $\alpha_{c}$ is a correction factor, $\rho_{0}$ is a characteristic distance, and $\mu_{c}$ is a parameter that controls how fast the force increases as the agents get closer. Note that the resultant force $F_{A}^{B}$ has speed dimensions. The normal and tangential vectors, $n_{A}^{B}$ and $t_{A}^{B}$, are calculated by:
(15a)$$\begin{array}{c} n_{A}^{B}=\frac{r_{A}^{B}}{|r_{A}^{B}|} \end{array}$$
(15b)$$\begin{array}{c} t_{A}^{B}=-\mathrm{sign}\left(V_{A}^{B}\cdot \left[-r_{Ay}^{B},r_{Ax}^{B}\right]\right)\cdot \frac{\left[-r_{Ay}^{B},r_{Ax}^{B}\right]}{|r_{A}^{B}|} \end{array}$$

Note that the virtual force over agent *B* due to the presence of agent *A* is exactly equal but with contrary direction, $F_{A}^{B}=-F_{B}^{A}$. If *N* agents are involved in the conflict, the total force is simply calculated as the summation of the virtual forces:(16)$$F_{A}=\sum_{n=1}^{N}F_{A}^{n}$$
where $F_{A}^{n}$ is the virtual collision force caused from the presence of agent *n* in agent *A*. When the total force is calculated, thanks to the conversion factor $\alpha_{c}$, it can be summed to the velocity command to correct it and avoid a collision:(17)$$V_{c}^{*}=V_{c}+F_{A}$$

Although this method does not guarantee that a group of agents reaches a deadlock situation, based on experience, it does not usually happen during the mission, even when the density of agents is very high. Anyway, in case of a deadlock, after some reasonable time elapsed without reaching the destination cell, the agents evaluate again the surrounding cells, selecting another one and solving then the deadlock. In Table 2, the selected values of the parameters of the virtual force equations have been shown.


*Cell exchange protocol*


In some situations, although the virtual force would avoid a collision between agents, their movements would imply separating too much from the trajectory between cells. In some of those cases, an exchange of the goal cells between the agents would solve the conflict without requiring wasting time and energy avoiding the collision. Take as example the situations shown in Figure 3a,b; if both agents exchanged their goal cells ($G_{A}$ and $G_{B}$), the collision would be avoided and both agents would be able to travel straight towards their new goal cells. Therefore, a cell exchange protocol would be beneficial for these cases.

Such a protocol should not relay on a complex communication system because keeping communications simple is key to preserve scalability of the complete swarm. Therefore, the cell exchange protocol must not be based on a negotiation method (with bidirectional communications) but on a individually decision taking. The proposed cell exchange mechanism can be divided into two different procedures: capture other agent’s cell and resolve coincident cells.

The first procedure is activated when an agent detects a potential collision in an early future with another agent. It then forecasts whether the collision would be avoided if both cells were exchanged. If it is so, then it automatically captures the other agent’s goal cell and broadcasts it. To avoid recursive exchanges if more than 2 agents are involved in the collision, each agent keeps a list of agents with which the cell has been exchanged in the past (*exchanged_agents_list*). In case the other agent’s identification number is in the list, the cell will not be captured, because it was already captured in the past. That list is cleared when a goal cell is reached. As it will be shown in Section 3.3, all the other agents are informed about the new and the old goal cells through the broadcast messages (that is, the agent informs about the new goal cell it is traveling to and the old cell that has been exchanged).

Note that when an agent *A* decides to capture the goal cell of agent *B*, this last one does not automatically realize that this has happened, although it may reach the conclusion by itself that it should capture the goal cell of agent *A* to avoid the collision (as agent *A* did). Anyway the agents must be provided with a mechanism to solve the situation in which two (or more) agents fly to the same cell.

Let us suppose that *A* decides to capture the goal cell of *B* and broadcasts the information once the decision has been taken. When *B* receives the message, it detects that both agents are flying to the same cell, and then *B* triggers the second procedure (resolve coincident cells procedure). First, *B* checks if *A* is in its own *exchanged_agents_list*. Since it is not, then checks that *A* is broadcasting an alternative cell (agent *A*’s old goal cell) and takes it as its new goal cell. Then *B* broadcasts the information, being the exchange of cells effectively done.

It could happen that once *A* has captured *B*’s goal cell, it processes its own information and realizes that it is sharing the same goal cell with *B* (*B* did not have enough time to execute the resolve coincident cells protocol). Agent *A* would then check if *B* is in its *exchanged_agents_list*, and given that it is indeed, would do nothing, waiting for *B* to take *A*’s old goal cell.

### 3.3. Communication Requirements

As above indicated, the surveillance algorithm is fully distributed and each agent executes each own control, updating its internal knowledge with the information sent by the other agents. Additionally, the collision avoidance procedure makes use of the position and velocity of the surrounding agents, as well as the goal and exchanged cells. Therefore, the needed information must be periodically broadcast by each agent so that the other agents receive it and update their internal knowledge. The message has the following fields:Time stamp.Status information, such as *taking off*, *landed*, *ready to start the mission*, battery level, etc.Identification number of the agent, ID.Current position, [x,y,z], in global axis.Current velocity, $\left[V_{x},V_{y},V_{z}\right]$, in global axis.Cell indexes, $\left[i_{x},i_{y}\right]$, to which the agent is heading to.Exchanged cell indexes, $\left[EC_{x},EC_{y}\right]$. If there is not currently any exchanged cell, this field is set to [−1,−1].

The frequency of the message broadcast has been set to 2 Hz. This frequency can be lowered without losing efficiency. However, given the small dimensions of the test bed and the low distances between the quadcopters, for safety reasons, it was decided to increase it up to that frequency.

### 3.4. Optimization

The complete control algorithm contains 13 parameters to be configured, whose optimal values depend on the scenario variables [17]: surveillance area per agent, $As/N_{a}$; number of agents, $N_{a}$; nominal speed, $V_{n}$; radius of the sensor footprint, $R_{f}$; and area shape factor, $f_{A}$. Then, for each tuple $\left(As/N_{a},N_{a},R_{f},V_{n},f_{a}\right)$, the algorithm must be optimized to find a set of values that maximizes the evaluation variable (in this case, the efficiency defined as per Equation (Equation 10)).

In past works [17], a genetic algorithm was used for the optimization of the former algorithm, showing good results compared with other methods, such as Bayesian optimization. Therefore, the same method has been used for this work with the following characteristics:Chain of genes: a vector made up by the 13 parameters of the algorithm. Each of the genes is normalized with a range of valid values.Population: 100 members.Initial population: randomly generated.Fitness function: To evaluate each member, Equation (Equation 10) is used. The duration of the surveillance has been set to 600 s. The efficiency is averaged over 5 simulations with different initial positions.Crossover: 50 new members are generated. The parents are paired using the roulette-wheel technique, with a probability proportional to the efficiency value. The genes of the parents are created by applying a weighted sum of each gene individually. The weighting coefficient is a random number between 0 and 1.Next generation selection: the new members are evaluated and the best 100 (from the total population of 100 parents and 50 of the offspring) are selected for the next generation.Stopping criteria: the optimization is stopped when one of these criteria is met:
-Maximum number of generations (20) has been reached.-Maximum number of generations (5) without an improvement higher than 10% of the best member has been reached.-Maximum number of generations (5) without an improvement higher than 10% of the mean efficiency of the population has been reached.

Note that, for this work, only 6 scenarios must be optimized (4, 6, and 8 agents, at 0.10 and 0.15 m/s), which needs a reasonable amount of time (The complete process took around 3 days in a personal computer. Considering 15 generations on average for each of the 6 scenarios, with 50 members to be evaluated and 5 simulations with different initial conditions each time, each simulation needs approximately 3·24·3600/(6·15·50·5) = 11.5 s. The time compression of the simulations is 600 s/11.5 s = 52.). However, one may argue that if the algorithm is to be used for a broad scenario type range (with different footprint radius or area size, for instance), the optimization process would become unfeasible in terms of computational time. For example, considering $R_{f}=$ 0.3, 0.6, 0.9 would multiply the number of optimizations by 3. In [17] it is explained in detail how this can be successfully done in a reasonable amount of time, so that the algorithm can adapt to any tuple $\left(As/N_{a},N_{a},R_{f},V_{n},f_{a}\right)$ within a previously known range of values. In Appendix A, a table with the final values of each of the 13 parameters have been presented for each scenario.

## 4. Simulation Results

Once the algorithm has been optimized for the 6 scenarios ($N_{a}=\left[4,6,8\right]$; $V_{n}=\left[0.10,0.15\right]$ m/s), it is tested 5 times during 600 s in each scenario changing the initial positions of the quadcopters. The efficiency is tracked over time as explained in Section 2.3. In Figure 4, a screenshot of the the mission after 300 s has been presented. Each agent (8 in total) is labeled with its identification number, and in background the age of each discretization cell is shown.

In Figure 5a,b the mean efficiency over time has been represented for different number of agents (4, in blue; 6 in red; 8 in green) and for the two speeds considered (0.10 m/s on the left figure and 0.15 m/s on the right one). For each case and with the same color, it has been included a band which represents the lower and higher efficiency found during the 5 simulations.

In the early stages of the missions, the efficiency tends to decrease until it reaches a lower bound before the first 150 s. After that point, the efficiency increases slowly for 4 and 6 drones, and stays constant in the case of 8 drones, reaching a steady efficiency around 0.6–0.5.

The most significant fact inferred from the results is that the efficiency decreases as the number of agents increases. From 4 to 6 drones, this drop is equal to 0.025 for $V_{n}=0.10$ m/s, and 0.030 for $V_{n}=0.15$ m/s. From 6 to 8 agents, the reduction is equal to 0.075 and 0.03 for 0.10 and 0.15 m/s respectively. This decrease in the efficiency as the concentration of agents (measured as the number of agents divided by the area) increases, has been also observed in past works. When the density of agents increases, the agents interfere between each other, making their movements more difficult.

Another interesting study is the contribution of each of the behaviors to the final decision. If the coincident cell behavior is not considered (since it does not encourage to select one cell, only prevents it), the contribution of each behavior can be calculated normalizing the absolute value of each term of the final decision equation with the sum of all the contributions. When a goal cell is selected, the total contribution is then calculated as:(18)$$S_{T}=|I_{P}\left|+\right|\alpha_{E}C_{E}\left|+\right|\alpha_{DM}I_{DM}\left|+\right|\alpha_{D}I_{D}\left|+\right|\alpha_{V}I_{V}|$$
where $S_{T}$ is the total contribution for the selected goal cell. The contribution of each behavior is finally calculated as:
(19a)$$\begin{array}{c} S_{P}=\left|I_{P}\right|/S_{T} \end{array}$$
(19b)$$\begin{array}{c} S_{E}=\left|\alpha_{E}C_{E}\right|/S_{T} \end{array}$$
(19c)$$\begin{array}{c} S_{DM}=\left|\alpha_{DM}I_{DM}\right|/S_{T} \end{array}$$
(19d)$$\begin{array}{c} S_{D}=\left|\alpha_{D}I_{D}\right|/S_{T} \end{array}$$
(19e)$$\begin{array}{c} S_{V}=\left|\alpha_{V}I_{V}\right|/S_{T} \end{array}$$
where *P* indicates the pheromone behavior, *E* the energy saving, DM the diagonal movements, *D* keep distance, and *V* keep velocity.

During the 5 simulations for each scenario case, every time a decision is taken by an agent, the contribution of each behavior is recorded. In Figure 6a–f, all the contributions have been represented for all the scenario cases. Going down through the rows of graphs, the number of drones increases; on the left column, the contributions for $V_{n}=$ 0.10 m/s; on the right, for $V_{n}=$ 0.15 m/s.

The pheromone behavior is the most important in most of the cases. Its contribution decreases as the number of agents increases, being even secondary for 8 agents and 0.10 m/s. With 4 agents and 0.10 m/s, this behavior is the only one relevant in the decision.

The contribution of the energy saving behavior is secondary in all the cases, although it becomes more important as the number of agents and the speed increase. Note that in the model, no energy consumption has been considered and therefore, the only benefit in using this behavior is to ease the travel between cells, since it encourages to keep the flying direction. For higher speeds, keeping the track becomes more difficult, which explains why this behavior contribution is higher for 0.15 m/s. It has been observed that often, regardless the scenario, the quadcopters trend to move diagonally. This can be explained with the combined contribution of the energy saving and diagonal behaviors.

In the case of the keep distance behavior, the contribution is very variable. For 4 drones, it is irrelevant. However, as the number of agents increases, it becomes more important, being the most relevant for 8 agents and 0.10 m/s, and similar as the pheromones behavior for 8 agents and 0.15 m/s. As the concentration of agents increases, it becomes more important to keep a distance between them to avoid unwanted interferences between them. This results reinforces the conclusion obtained observing the efficiency: as the number of agents increase, it becomes more difficult to get a high efficiency.

Finally, the velocity behavior does not contribute somewhat in the final decision. This behavior is designed to encourage the agents to fly to areas following other quadcopters. However, given that the surveillance area is not big, this behavior does not provide much benefit.

## 5. Robustness Analysis

### 5.1. Lost of Broadcast Messages

The first test to verify the robustness of the algorithm is the loss of broadcast messages. Each agent broadcasts information at 2 Hz, as indicated in Section 3.3. In a real world scenario, depending on several factors such as complexity of the scenario, number of agents or weather conditions, some messages may be lost and some of the agents would not be able to update their own internal information. Based on a limited knowledge, it is expected that the decisions taken are less efficient and the overall algorithm performance is consequently impacted.

To test this, each agent will individually dismiss a received message depending on a probability $P_{l}$. Each scenario is then tested 5 times and its final efficiency after 600 s is calculated, averaged, and normalized with the efficiency without any message loss, i.e., $P_{l}=0$. In Figure 7a,b the normalized efficiency has been plotted depending on the probability of loosing a message $P_{l}$ for the different scenarios.

As it can be seen, the normalized efficiency stays close to 1 up to $P_{l}=0.8$; at that point, the efficiency suffers a very important drop. For $P_{l}=1$, no interaction between the agents takes place, that is, each agent acts individually and the swarm does not cooperate. In that case, the normalized efficiency drops to ∼0.65 (∼65% of the efficiency with full communications) for 4 drones, to ∼0.50 for 6 drones, and to ∼0.45 for 8 drones. As expected, the higher the number of drones, the more needed the communication is to achieve a high efficiency.

### 5.2. Positioning Errors

Given that the quadcopters usually estimate their position in outdoor environments using the Global Position System (GPS), they are subjected to errors and noises when locating themselves in the space. The GPS is also used to estimate the velocity, although those errors are negligible [21]. To take this into account, the impact on the efficiency due to the positioning errors is investigated.

Knowing that the scenario under study is a small representation of its counterpart in the real open-space world (smaller area and sensor footprint, and slower speeds), the same error magnitudes cannot be considered. For example, a typical GPS positioning error could be 7 m, which cannot be directly applied to the here studied indoor case. Therefore, the positioning errors must be scaled down using a factor.

The speed of the quadcopters considered in this work (for the indoor arena) has been 0.10 and 0.15 m/s. A bigger-size quadcopter, designed for outdoor surveillance missions, could be assumed to be around between 5 and 10 m/s. A valid scale factor, under this hypothesis, could be then $0.10\;{\mathrm{ms}}^{-1}/5\;{\mathrm{ms}}^{-1}=0.02$. This means that, if a positioning error of 0.20 m is considered in this arena, in the real open-space world the equivalent error would be 0.20/0.02=10 m. Similarly, the quadcopter would fly in the real world at 0.10/0.02 = 5 m/s, and 0.15/0.02 = 7.5 m/s, and would have a sensor footprint with a radius of 0.30/0.02 = 15 m. (Although all these converted values are sensible, different scaling factors can be considered. For instance, if we know that the sensor footprint in the real world has a radius of 50 m, the scale factor will be 0.30/50 = 0.006. Results shown in this section should be accordingly reinterpreted with this new value.)

The positioning error proposed for this small-scale scenario is calculated as follows, similarly as in [22]. Having defined an error $X_{e}$, the positioning error is calculated and updated with a frequency of 1 Hz as per:
(20a)$$\begin{array}{c} R_{e}∼N\left(X_{e},{\left(X_{e}/4\right)}^{2}\right) \end{array}$$
(20b)$$\begin{array}{c} \theta_{e}∼U\left(0,2\pi \right) \end{array}$$
where N and U indicate Gaussian and uniform distributions. $R_{e}$ is the distance from the real to the measured position, and $\theta_{e}$ is the direction of the error. The measured position, including the error, is then obtained:
(21a)$$\begin{array}{c} x_{m}=x_{r}+R_{e}\cos\left(\theta_{e}\right) \end{array}$$
(21b)$$\begin{array}{c} y_{m}=y_{r}+R_{e}\sin\left(\theta_{e}\right) \end{array}$$
where the subscript *m* indicates measured and *r* indicates real.

Depending on the value of $X_{e}$, the algorithm has been tested 5 times for the different scenarios, measuring the final efficiency after 600 s. These efficiencies have been averaged and again normalized with the mean efficiency without positioning error. In Figure 8a,b, the normalized results have been shown. As it can been seen, there is a linear loss of efficiency, more relevant when the number of drones is higher, and it is independent from the flying speed. Considering an error equal to the size of the sensor footprint, there is a loss of efficiency between 16% and 22%. Recall that considering a resize factor of 0.02, that positioning error in a real world scenario would be around 15 m, quite big considering the GPS.

### 5.3. Failure of Drones

The third and final robustness test is to explore how the aerial swarm would reconfigure itself if some of the agents eventually stop working. Once the surveillance task has been started, half of the agents are forced to land after 100 s, and they do not participate in the mission anymore.

In Figure 9a,b, the efficiency during the mission, with a forced land of half of the quadcopters has been presented. Again, the efficiency has been averaged with 5 simulations, and normalized with the efficiency without failures. To see the complete evolution of the efficiency in the long term, the simulations in this test have been extended up to 4000 s. During the first 100 s, the normalized efficiency remains close to 1, as expected, since no failure has taken place yet. After the failure (marked in the graphs with a black dashed line), the efficiency immediately starts decreasing, stabilizing after t= 1000 s. Note that, given that half of the agents stops participating in the mission, the expected final normalized efficiency should drop down to 0.5 (%50 of the efficiency if all the agents were available). However, the efficiency stays in the long term above that value. This may be because although the algorithm is not configured with optimal values for the new number of agents, the density of agents in the area is lower, which helps to reach a higher efficiency. This fact would also explain why the curve of 8 drones remains over the other two.

## 6. Experiment Results

To analyze the reality gap between simulations and real robots, and to validate the implementation of the algorithm on a real system, experiments were carried out in a test bed. Due to the possibility of collision between the quadcopters, tests with loss of broadcast messages and induced positioning errors were not performed. Moreover, since the experiments require a considerable amount of time, tests with $N_{a}=$ 6 and 8 drones for a $V_{n}=$ 0.15 m/s were not carried out.

### 6.1. Test Arena and Drones Used

The tests were carried out at Kyoto University, Matsuno Laboratory, in the arena shown in Figure 10a. The inner dimensions of the usable volume are 6.4×4.0×3.0 m, and the motion of the quadcopters is captured by 10 Optitrack Prime 17W cameras located around the perimeter. The drones used are 8 Parrot Mambo Minidrones quadcopters (see Figure 10b), with a weight around 70 g and a typical autonomy between 250 and 500 seconds, depending on the battery capacity and wear (two battery types, 550 and 700 mAh), and flying speed. Infrared reflectors were attached in different positions on the drones to allow the capture system to track them.

In Figure 11, a general scheme of the arrangement of the different elements of the experiment setup has been represented. The motion capture cameras are connected to a switch, to which a computer running Motive Optitrack (Natural Point) is connected as well. This computer receives the images from the cameras, processes them and makes the position of the quadcopters available at a TCP/IP network. A second computer receives the position of each agent and calculates their velocity, acceleration and heading. In this very computer, the agents’ intelligence is executed, and the commands are sent to the Minidrones by Bluetooth (two computers needed for more that 7 devices). A supervisor agent runs in this computer as well, which is in charge of monitoring the task, send high level commands (such as take-off, land and start the task) and save telemetry data. All the system has been implemented in the Robot Operating System (ROS).

At the beginning of each test, the drones are landed in random positions. The supervisor sends a take-off command and when all the drones reach the flying altitude, the supervisor sends a command to start the surveillance. When one of the drones runs out of battery, a command is sent to land the rest of them, and the test is finished. A video with some test examples is available on-line (https://youtu.be/zUetTkswOkE).

### 6.2. Efficiency

The first of the two tests carried out is the efficiency tracking of the algorithm during a simple mission, similar to the one analyzed in Section 4. With 4, 6 and 8 quadcopters and a nominal speed of 0.10 m/s, and with 4 drones and a nominal speed of 0.15 m/s, 5 tests were performed and their efficiency tracked.

In Figure 12a,b, the efficiency of the considered cases has been represented, averaged for the 5 tests. The colored band associated with each case represents the interval of minimum and maximum efficiency registered. As it can be seen, the duration of the missions are shorter than in the simulations. Also, with a higher number of quadcopters, the duration is lower because the mission finishes when one of them runs out of battery, and the higher the number of drones, the higher the probability is that this happens prematurely. Note also the abrupt variations either in the mean and in the interval bands of the efficiency, indicating that at those times one of the tests finishes (for example, in Figure 12a, for $N_{a}=$ 4, at t= 500 s, one of the experiments finishes and the blue-colored band disappears, indicating that only one of the tests endured that long.)

The behavior of the efficiency is very similar to the one observed in the simulations. In the first 100 s, the efficiency decreases and then starts increasing slowly. The final values lie between 0.55 and 0.65, as happened in the simulations. The efficiency turns out to be higher for lower number of agents (as observed in the simulations), and there is no relevant differences between the two nominal speeds considered for $N_{a}=$ 4.

To better compare the results of the simulations and the experiments, in Table 3 the efficiency of the algorithm, alongside with the interval of maximum and minimum efficiencies found, has been collected in 5 different moments during the mission. As it can be seen, there is a coincidence between simulations and experiments in most of the instants. Note that for some experiments, no interval or even no data is available because of the limited duration of the batteries.

### 6.3. Failure Test

The second experiment carried out to validate the result of the simulations has been the tracking of the efficiency if half of the agents stop working at certain point of the mission (t= 100 s), in the same way as it was done in Section 5.3. In Figure 13a,b, the efficiency tracked over time for 4, 6 and 8 agents and $V_{n}=$ 0.10 m/s, and for 4 agents and $V_{n}=$ 0.15 m/s, has been shown. As done before, the efficiency has been normalized with the efficiency without any failure, $E_{0}$. After 100 seconds, half of the drones are forced to land (marked with a dashed line in the graphs), and as a consequence, the efficiency starts decreasing gradually. In the first 100 s, before the failure, the normalized efficiency remains close to 1, as expected. The efficiency drop is very similar for 4 and 6 drones, and it is independent of the nominal speed.

Given that the duration of the batteries is very limited, a long-term behavior can not be observed in the experiments. However, in Table 4, the values of the normalized efficiency and the interval can be compared in simulations and experiments. As in the previous test, no big differences between simulations and experiments are observed.

## 7. Case Study: Surveillance Mission with Intruders

Let us suppose we are in charge of controlling a certain area of a sea close to our coast. In that portion of water, there may be different ships passing through it; some of them will be considered as friendly (such as fishing ships or pleasure boats), and others as enemies (for example, drug trafficking boats). Our mission is continuously controlling the area, flying over it at certain altitude with drones equipped with cameras. When an object is detected, to determine whether it is friend or enemy, the agent reduces the altitude to better observe it. After some needed time, if the object is categorized as friend, the agent recovers its nominal altitude and keeps on with the surveillance mission. If an enemy is recognized, the agent communicates it to the coast guards and tracks it until they intercept it.

Inspired by the above situation, the following scenario with real robots is proposed to test the algorithm:4 quadcopters continuously fly over the area (6.4 × 4 m${}^{2}$), looking for possible intruders. The flying speed is 0.10 m/s or 0.15 m/s.3 robots (see Figure 10b), representing intruders, move continuously on the ground with nominal speeds between 0 and 0.10 m/s. Each ground robot generates a random point, to which they move avoiding collisions between them. When a robot reaches its destination point, it generates a new one, and moves towards it again.Each ground robot belongs to one of two types: friend or enemy. Each ground robot generates its type every 60 seconds with the same probability of being friend or enemy.When a quadcopter detects one of the robots, it reduces its altitude from 1.5 to 0.5 m to observe it. The quadcopter has the ability of discerning whether the robot is friend or enemy in 30 s. If the robot is friend, the quadcopters flies back to the nominal altitude and keeps on with the surveillance. If it is declared as enemy, the quadcopter tracks it for another 30 s.While an intruder is being observed, it does not change its type. When it is not being observed anymore, after 30 s, it generates another type.

For each quadcopter and robot speeds considered, 5 missions are tested, tracking the surveillance efficiency and counting how many times the intruder types have been detected, recording the total number of intruder types that have been during the mission. In Figure 14a,b, an instant of the mission has been presented. Each agent is identified by a number, written in both figures. The intruders (yellow ground robots) are colored in the left figure with black if they are friendly, and in red if they are enemies. As it can be seen, the quadcopter number 2 is observing in that moment one of the intruders, recognized as enemy, while the others keep on with the surveillance task.

In Table 5, an overview of the results of the case study considered has been shown. For each of the scenarios, defined by the nominal speed of the quadcopter ($V_{n}$) and the intruders ($V_{n_{r}}$), the final efficiency, calculated over the 5 tests, has been represented. Additionally, it has been indicated the percentage of the intruders detected during all the tests, discerning between friends and enemies. As it can be seen, the efficiency stays close to nominal values (around 0.60, see Figure 12a,b), while around 35% of the intruders are detected.

## 8. Conclusions and Future Works

In this work, a behavior-based algorithm to carry out a surveillance task has been presented. The algorithm is based on a former one, originally designed for search missions. Simulation results have shown a competitive efficiency, while being robust against three different factors: loss of messages, positioning errors and failure of agents. Experiments have been also carried out to support simulation results in an indoor arena with small quadcopters. Finally, a case study has been presented to show a realistic implementation of the algorithm with both aerial and ground robots. When one of the quadcopters has to stop with the surveillance task (in that case, when an intruder must be observed and tracked), the algorithm allows the rest of the group to keep on with the surveillance.

The algorithm has been evaluated with an absolute and objective efficiency measurement. The efficiency reaches values around 60%, and suffers a decrease as the number of drones increases. Simulation results in terms of efficiency have been shown to be very similar to the experiments. The use of each of the behaviors has been analyzed as well, showing that the pheromones and the keep distance behaviors are the most used ones for the scenarios considered. When analyzing the robustness of the method against loses of messages, the algorithm keeps nominal efficiencies up to 80% of messages lost. When considering positioning errors, the efficiency decreases linearly; for an error $X_{e}=0.30$ m, (similar to 15 m in an open-world scenario), the algorithm loses between 16% and 22% of efficiency. Finally, if half of the agents stop working, the efficiency losses less than a 50%. In the proposed case study, the fraction of detected intruders has been tracked for different cases, reaching between 19% and 44%.

Considering the simulation and experimental results, and based on previous works, we claim that the proposed approach has demonstrated to be:Distributed: the algorithm can be executed on board, using the information broadcast from the other agents. Only very high level commands are needed from the central control (such as start or finish the mission).Robust: against losses of messages between the agents, positioning errors, and failures of some members of the team.Flexible: it can adapt to different scenarios (different number of agents, area size, flying speeds, etc.), keeping a high efficiency. Areas with higher interest can be included, as well as obstacles.Stochasticity: which may be important in some cases, given the difficulty in forecasting the agents’ movements, and making more difficult to burst into a sensitive areas without being detected.

On the contrary, the algorithm presents two main drawbacks. First, finding optimal values for any scenario is not a trivial task, and its implementation is complex. Secondly, in its present form, requires that all the agents have the same sensor footprint.

In future works, some topics can be tackled. When one or more agents leave the group for any reason (failure, return to base, assigned to other task, etc.), the algorithm could be automatically reconfigured to the new number of available agents. As it has been done in previous works [16], obstacles inside the area, or zones with more interest, can be considered. Regarding the communications, more detailed studies will be carried out; for instance, analyzing the restriction of the communication range, or considering the probability of loosing messages depending on the orography. More intelligent intruders can be included, which try to attack sensitive areas; the quadcopters would have to detect them before they reach those areas. Finally, different area shapes will be also studied, as well as the possibility of changing the flying altitude.

## Figures and Tables

**Figure 1 sensors-19-04584-f001:**
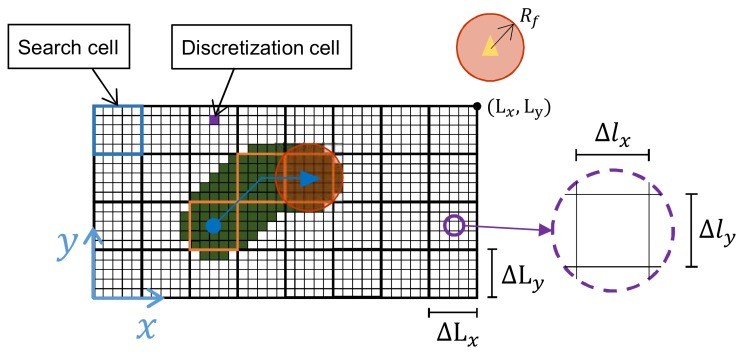
Scheme of the scenario considered. Each quadcopter has a sensor footprint with a radius $R_{f}$. The area, with lengths $L_{x}$ = 6.4 m and $L_{y}$ = 4.0 m, is divided into search and discretization cells, in blue and purple respectively. In the figure, the agents has flown between the three centers of the orange-marked cells. In dark green, the observed discretization cells during the movement.

**Figure 2 sensors-19-04584-f002:**
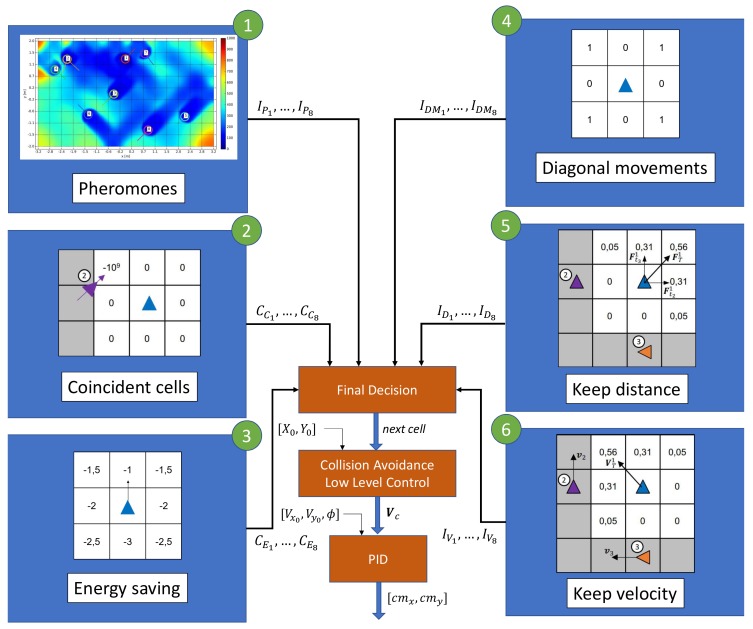
Scheme of the complete control algorithm of the agents. There are 6 behaviors: pheromones-based (1), coincidence goal cells avoidance (2), energy saving (3), diagonal movement (4), keep distance (5), and keep velocity (6). Each output is fed to a final decision module, which applies a weighted addition, selecting the next cell to head to. After being the goal cell selected, a low level control, which includes a collision avoidance algorithm, generates the velocity command referenced to the inertial frame, $V_{c}$. This command is passed to a PID controller, that finally generates the command in the body frame of the quadcopter, $\left[cm_{x},cm_{y}\right]$, see Equations (Section 2.2).

**Figure 3 sensors-19-04584-f003:**
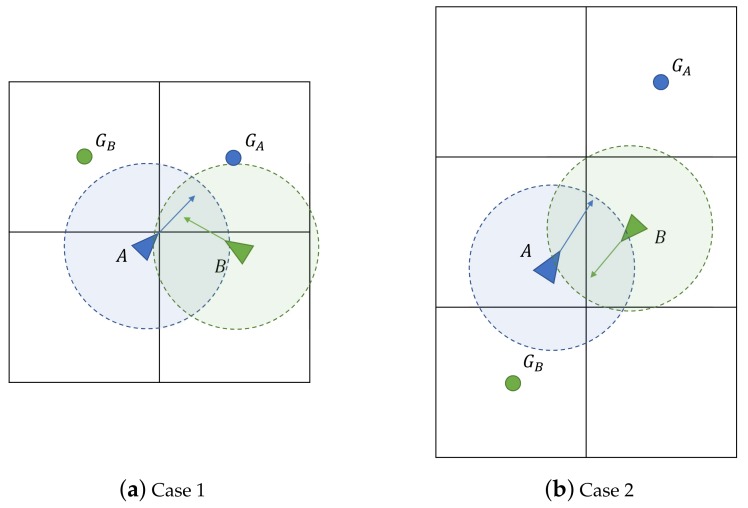
Examples of situations in which a collision may take place. Agents *A* (blue) and *B* (green) head to their respective goal cell, $G_{A}$ and $G_{B}$. A safety region with a radius $R_{safety}$ has been drawn surrounding each agent with the color of the agent that belongs to.

**Figure 4 sensors-19-04584-f004:**
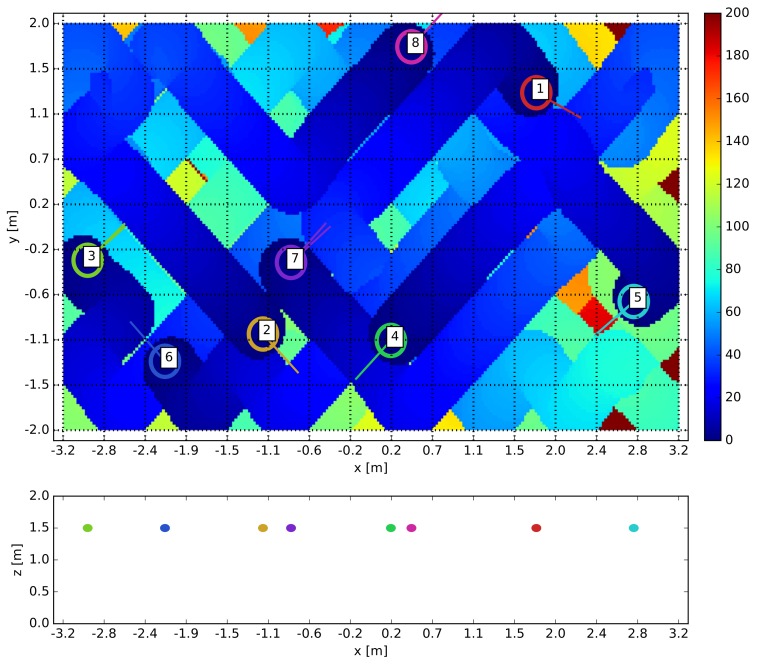
Example of surveillance mission with 8 drones and a nominal speed of 0.10 m/s after 300 s. The colored background represents the age of each cell. In the plot below, the position of the agents in the xz plane has been represented.

**Figure 5 sensors-19-04584-f005:**
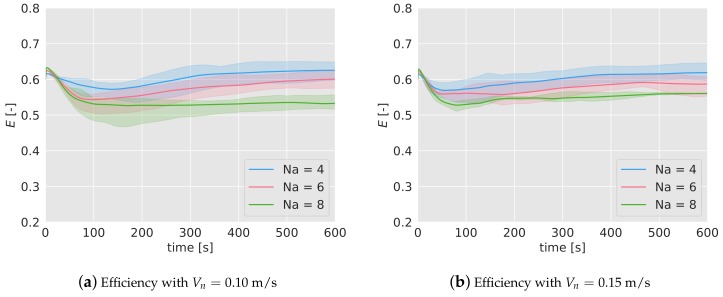
Surveillance mean efficiency, calculated from 5 simulations, during the mission with different number of drones, $N_{a}=4,6,8$. In (**a**) with a nominal speed of 0.10 m/s; in (**b**), for 0.15 m/s. The colored bands indicate the range within the efficiency lies for the 5 simulations carried out.

**Figure 6 sensors-19-04584-f006:**
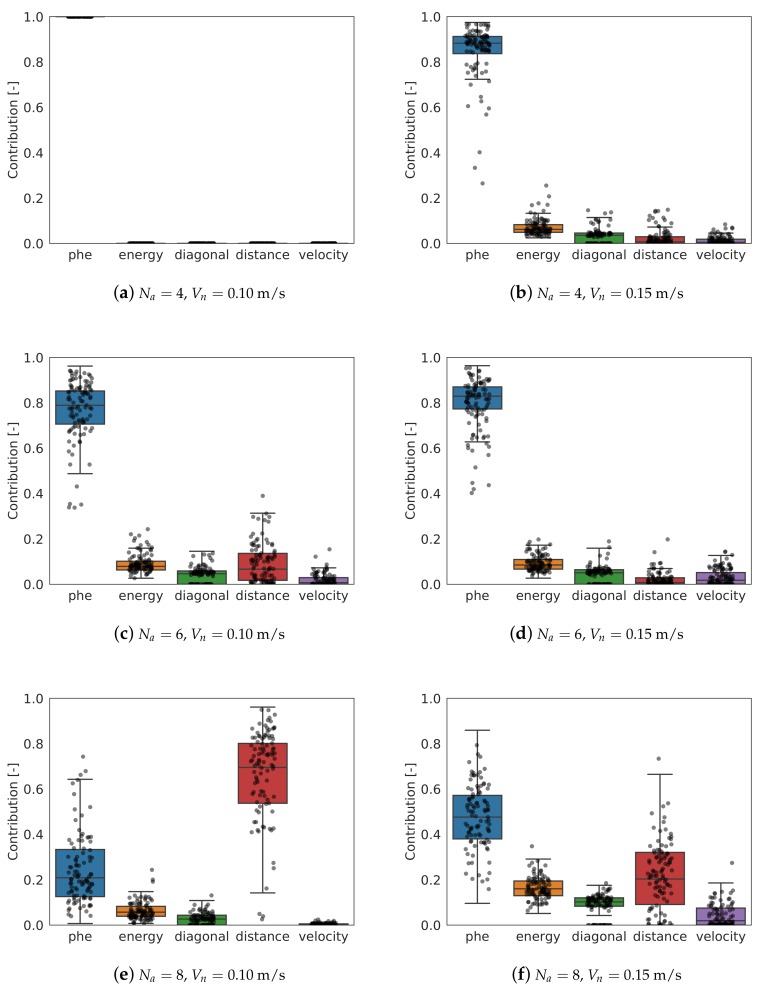
Normalized contribution of each behavior to the final decision as per Equations (Equation 18) and ([Disp-formula FD19a-sensors-19-04584]). The data has been extracted from 5 simulations and each decision taken by each agent. Some data (black dots) has been plotted as well.

**Figure 7 sensors-19-04584-f007:**
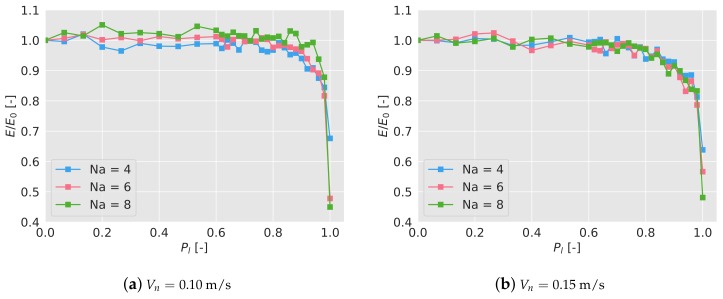
Efficiency of the algorithm depending on the probability of losing a broadcast efficiency, normalized with the efficiency with no lost messages, $E_{0}$. The messages are broadcast at 2 Hz.

**Figure 8 sensors-19-04584-f008:**
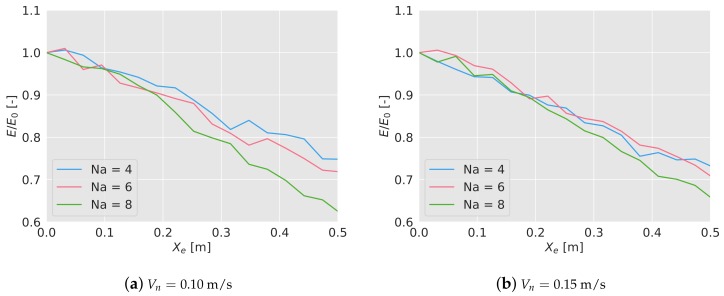
Efficiency of the algorithm depending on the positioning error $X_{e}$, normalized with the efficiency with no error, $E_{0}$. As the error increases, the algorithm loses efficiency.

**Figure 9 sensors-19-04584-f009:**
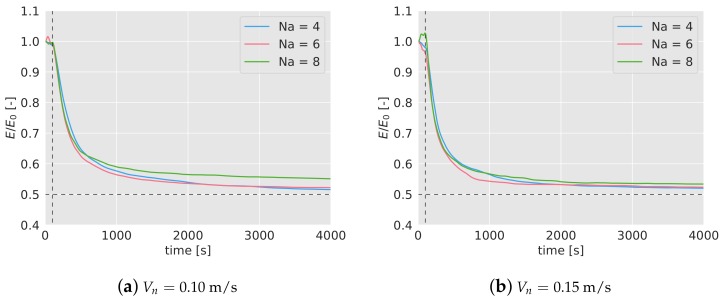
Efficiency of the algorithm over time when half of the agents are forced to land at t= 100 s, marked with a black dashed line. Theoretically, when half of the agents stop working, the efficiency is expected to drop down to 0.5, marked with a black dashed line as well. The efficiency has been normalized with the efficiency without quadcopter failures, $E_{0}$.

**Figure 10 sensors-19-04584-f010:**
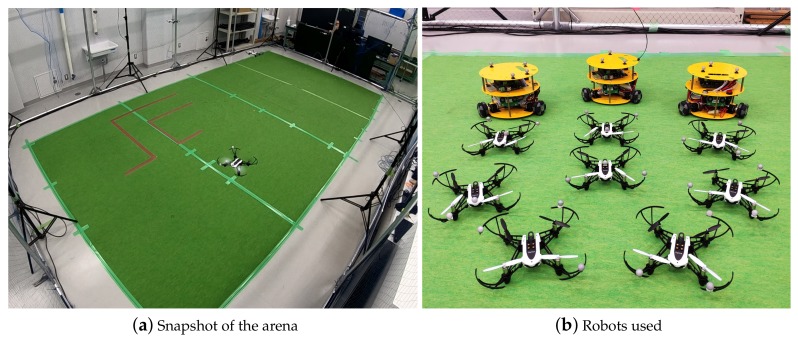
Test arena and robots used. In (**a**), a snapshot of the arena during a surveillance test. In (**b**), the quadcopters used (Parrot Mambo Minidrones). In the same figure appear the ground robots that were used for the case study mission, as it will be presented in Section 7.

**Figure 11 sensors-19-04584-f011:**
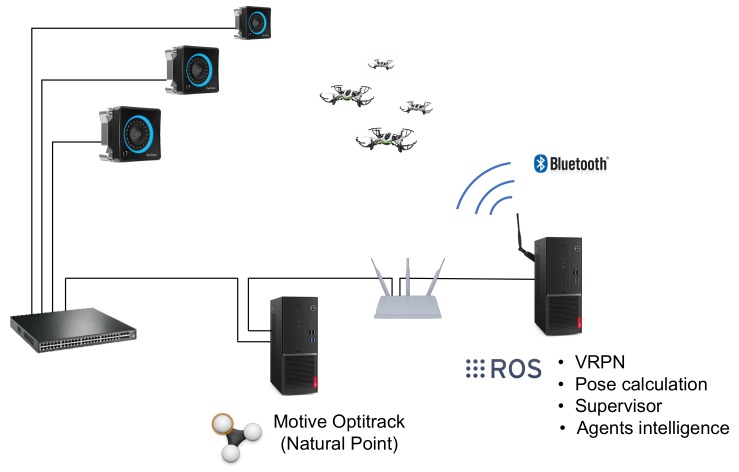
General scheme of the experiment’s setup.

**Figure 12 sensors-19-04584-f012:**
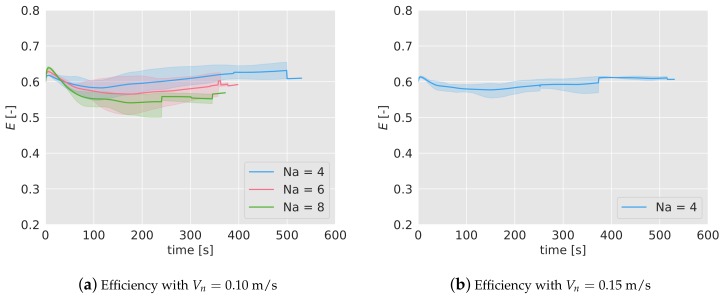
Surveillance efficiency during the experiments with different number of drones, $N_{a}=4,6,8$. In (**a**) with a nominal speed of 0.10 m/s; in (**b**), for 0.15 m/s with 4 drones. The colored bands indicate the range within the efficiency lies for the experiments carried out.

**Figure 13 sensors-19-04584-f013:**
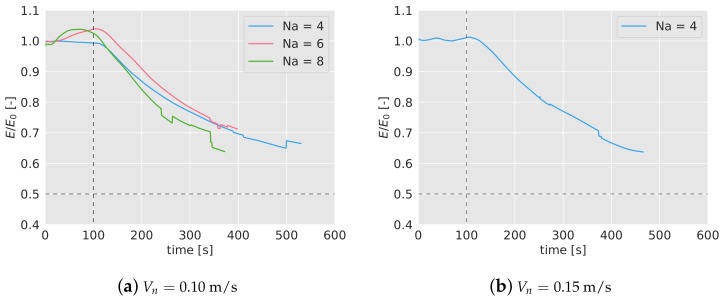
Efficiency of the algorithm over time when half of the agents are forced to land at t= 100 s during the experiments, marked with a black dashed line. The efficiency has been normalized with the efficiency without quadcopter failures, $E_{0}$. Note that the lines are uneven due to the different durations of the experiments.

**Figure 14 sensors-19-04584-f014:**
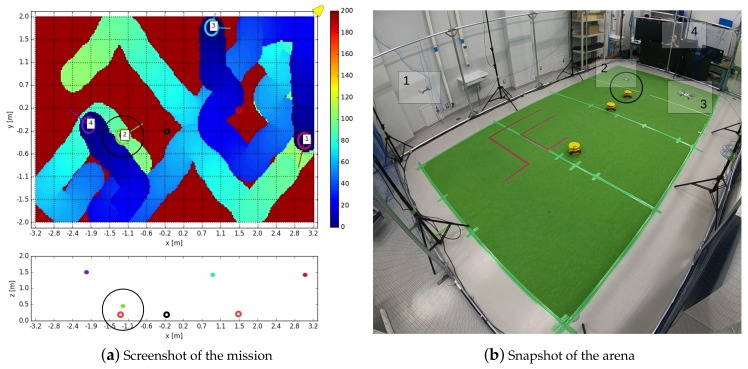
Screenshot of the surveillance mission at a certain instant. In Figure 14a, in the upper graph, the position of each quadcopter and ground robots in the xy plane has been represented. In the background, the age of the cells. Each of the 4 agents has been labeled with its ID. The position on the camera has been indicated with a yellow trapezoid. In the lower graph, the position in the xz plane. Friendly intruders have been plotted in black; enemy intruders in red. Note that agent number 2 is observing (flying at a low altitude) one of the enemy intruders. In Figure 14b, a snapshot of the arena.

**Table 1 sensors-19-04584-t001:** Values of the parameters of the potential field implemented to avoid collisions.

$b_{x}$ [m/s${}^{2}$]	$b_{y}$ [m/s${}^{2}$]	$T_{d}$ [s]	$c_{x}$ [s${}^{-1}$]	$c_{y}$ [s${}^{-1}$]
8.45 ± 0.37	7.34 ± 0.24	0.25	0.28 ± 0.02	0.26 ± 0.02

**Table 2 sensors-19-04584-t002:** Values of the parameters of the virtual force implemented to avoid collisions.

$\alpha_{c}$ [-]	$\rho_{0}$ [m]	$\mu_{c}$ [-]
10	0.6	2

**Table 3 sensors-19-04584-t003:** Comparison of the efficiencies registered in simulations and experiments, obtained from Figure 5a,b and Figure 12a,b. *Sim.* indicates simulation and *Exp.* experiment. The values outside the brackets are the mean values, whereas the values inside them are the interval of maximum and minimum efficiencies found.

		$V_{n}=$ 0.10 m/s			$V_{n}=0.15$ m/s
		$N_{a}=4$	$N_{a}=6$	$N_{a}=8$	$N_{a}=4$
t=100 s	Sim.	0.58 [0.55, 0.60]	0.54 [0.51, 0.56]	0.53 [0.49, 0.55]	0.57 [0.53, 0.60]
	Exp.	0.58 [0.55, 0.60]	0.57 [0.54, 0.61]	0.55 [0.53, 0.57]	0.58 [0.57, 0.59]
t=200 s	Sim.	0.58 [0.54, 0.60]	0.56 [0.51, 0.58]	0.53 [0.48, 0.55]	0.59 [0.56, 0.62]
	Exp.	0.60 [0.57, 0.63]	0.57 [0.52, 0.62]	0.54 [0.50, 0.57]	0.58 [0.57, 0.60]
t=300 s	Sim.	0.61 [0.56, 0.64]	0.58 [0.54, 0.60]	0.53 [0.50, 0.54]	0.60 [0.58, 0.62]
	Exp.	0.61 [0.59, 0.64]	0.58 [0.55, 0.62]	0.55 [0.54, 0.56]	0.59 [0.58, 0.61]
t=400 s	Sim.	0.62 [0.59, 0.65]	0.58 [0.56, 0.61]	0.53 [0.51, 0.55]	0.61 [0.59, 0.64]
	Exp.	0.63 [0.61, 0.65]	0.59 [-]	-	0.61 [0.61, 0.61]
t=500 s	Sim.	0.62 [0.59, 0.65]	0.60 [0.58, 0.62]	0.54 [0.52, 0.55]	0.62 [0.59, 0.65]
	Exp.	0.63 [0.61, 0.65]	-	-	0.61 [0.61, 0.61]

**Table 4 sensors-19-04584-t004:** Comparison of the efficiencies registered in simulations and experiments, obtained from Figure 5a,b and Figure 12a,b. *Sim.* indicates simulation and *Exp.* experiment. The values outside the brackets are the mean values, whereas the values inside them are the interval of maximum and minimum efficiencies found. Note that for some experiments, no interval or even no data is available because of the limited duration of the batteries.

		$V_{n}=$ 0.10 m/s			$V_{n}=0.15$ m/s
		$N_{a}=4$	$N_{a}=6$	$N_{a}=8$	$N_{a}=4$
t=100 s	Sim.	0.99	0.99	1.00	0.98
	Exp.	0.99	1.03	1.04	1.01
t=200 s	Sim.	0.88	0.85	0.85	0.83
	Exp.	0.87	0.91	0.84	0.88
t=300 s	Sim.	0.77	0.72	0.73	0.71
	Exp.	0.77	0.78	0.72	0.77
t=400 s	Sim.	0.69	0.66	0.67	0.65
	Exp.	0.70	0.71	-	0.67
t=500 s	Sim.	0.65	0.63	0.64	0.62
	Exp.	0.67	-	-	0.64

**Table 5 sensors-19-04584-t005:** Overview of the case study results, considering 7 different scenarios (from $S_{1}$ to $S_{2}$). $V_{n}$ is the nominal speed of the quadcopter; $V_{n_{r}}$, the speed of the ground robots; $\bar{E}$ the mean efficiency for each scenario over 5 tests; *Detected friendly*, the percentage of friendly intruders that have been detected; *Detected enemy*, the percentage of enemies detected.

Scenario Number	$S_{1}$	$S_{2}$	$S_{3}$	$S_{4}$	$S_{5}$	$S_{6}$	$S_{7}$
$V_{n}$ [m/s]	0.1	0.1	0.1	0.1	0.15	0.15	0.15
$V_{n_{r}}$ [m/s]	0.00	0.03	0.05	0.08	0.03	0.06	0.1
$\bar{E}$ [-]	0.54	0.54	0.55	0.53	0.53	0.50	0.52
Detected friendly	36%	20%	19%	43%	37%	44%	26%
Detected enemy	27%	32%	28%	33%	38%	31%	35%

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
