# Peer review of "Behavior-Based Control for an Aerial Robotic Swarm in Surveillance Missions"

_sensors, 2019, doi:10.3390/s19204584_

Round 1

Reviewer 1 Report

Pleas see uploaded file called "Reviewer Report.pdf".

Author Response

We would like first to thank the reviewer for the comments provided. The comments have been here collected and answered one by one. Major changes have been highlighted. A link to a uploaded video has been added in page 20.

We note that the dimensions of the terrain used for the tests is rather small compared to the surface seen by a drone, which results in the drones spending much of their time avoiding collisions. It would definitely be worthwhile for the authors to pursue their experiments with a larger terrain. Nevertheless, the experiments are well conducted and the results are very interesting.

As explained in Section 2.1 and 2.2, the dimensions of the arena (terrain) are fixed (dimensions of the test bed), whereas the dimension of the sensor footprint has a low boundary value related to the size of the physical quadcopters, Parrot Mambo Minidrone. The algorithm can be used in larger areas though. In the future, we are planning to carry out experiments with larger drones in open spaces, where the ratio terrain size / quadcopter size will be much larger.

Everywhere: the authors should replace "velocity" with "speed" since the velocity of an object is a vector while its speed is the magnitude of its velocity.

Velocity has been replaced by speed when referring to velocity's magnitude

 P. 2: First paragraph: the authors should give some references for both the ground robots and the UAVs.

Please, note that reference [3] surveys tasks with ground robots, which anyway is not the main focus of this work. In sections 1.2 and 1.3 past works regarding UAV swarms are studied in detail.

P.3: Section 1.4: The authors should state explicitly the difference between an algorithm designed for searching and one designed for surveillance. This would help the reader better appreciate what the authors have done.

A paragraph has been added indicating the main two differences between both algorithms. Note that in Sections 2.3 and 3.1 the differences have been detailed more clearly.

P. 5: Eqs 2-3: the authors should say a word about the origin of these equations, mainly: why these transfer functions.

The equation 2 is a first-order time-delay system, as specified in the previous paragraph. Equation 3 transforms the commands into the system inputs.

Some clarification has been added.

P.5 Change Rsafety to 30 cm instead of 0.30 cm.

Similarly, in Eqs. (4a), (4b), and the following line, should it not be m instead of cm ?

Dimensions have been set to meters (m)

P. 7, After Eq. (10): About the discussion of the initialization of the age of the cells.

It seems that logically, the initial age of the cells should be set to infmity because they have never been seen already. Thus, the initial efficiency would be zero, which is correct since no surveillance has yet occurred. As the agents travel through the region considered, the age of the cells would decrease and the efficiency increase as should be expected. The authors should explain why this would be wrong.

This is a very interesting reflection.

If the initial age is set to infinity, the mean age of the information would be also infinity until all the cells have been observed at least once. Infinity mean age implies zero efficiency. Thus, until all the cells have been observed once, the efficiency would be zero. Given that the final efficiency is an average over time, it would be greatly affected by the solely decision of setting the initial age. During the optimization, the best solutions would probably be the ones that enforce the rapid observation of all cells, which is not the aim of surveillance (which actually is having updated information).

Selecting an initial age so that the initial efficiency is equal to the final expected one is the best way to interfere as low as possible in efficiency measurement.

If the mission would last a very long time, it would not really matter much because the average would be over a lot of time and the initial moments would have a low weight. However, we have only considered 600 seconds due to the batteries capacity.

Some further details have been given for clarification.

Furthermore, in practical applications, at the beginning of a mission, it seems very improbable that one would know the fmal efficiency that is required in order to use the definition of Eq.(11). It seems to me that a defmition independent of "the fmal efficiency" should be given.

As we have demonstrated with the experiments, simulation results turn out to be very accurate. This means that given a scenario (number of drones, dimension of the area, speed of quadcopter, etc.) we can know in advance the expected efficiency. Anyway, even if the final efficiency turns out to be very different to the forecast, selecting 0.6 is going to be closer to the real one than setting the age to infinity (efficiency equal to 0).

P.7, 8, 9: The description of the algorithm is well done.

Thanks.

Line 342: Based on the experience of our group with similar computations using the genetic algorithm, I would say that the computation time reported is much too long for the task at hand. The authors should explain why it takes so much time.

A lot of simulations have to be carried out for the complete optimization (6 scenarios, 15 generations of average, 50 new members each generation, 5 different simulations to calculate the efficiency on average). Simulation turns out to have a compression time of 52, which is quite fast (simulator implemented in Python).

A footnote has been added.

Figure 4 shows that, with so many agents in such a small region, most of the time is spent avoiding collisions with other agents, instead of surveying the region. In terms of surveillance, this would be rather inefficient. This is adequately pointed out by the authors at P. 13, Lines 367­-368 and at P. 15, Lines 389-390. Although it is not necessary for the authors to do so in this article, it would be interesting for them to develop a method to determine what the ideal/best number of agents would be for a given situation.

In terms of age of the information, the higher the number of drones, the more updated the information is. Therefore, the ideal number of drones is the highest available.

In terms of efficiency, with this size of the area and size of drones, there is a decrease in the efficiency as the number of drones increase. Extrapolating the results, 1 drone would provide the highest efficiency (although with 1, 2 and 3 drones it has not been tested, it is just an hypothesis).

This will be further studied in the future.

P. 13, Section 4: The discussion of the results is well done.

Thanks

P. 17, Line 410 should be changed to "... the swarm does not cooperate..."

Changed, thanks.

P. 17, Line 423: How is this scaling to "the real world" obtained? ... What is the definition of "the real world"?

Further clarification has been added in the section. We hope it's more clear now.

P .24 L 550: "a competitive efficiency" ... as compared to what?

In past works, algorithms have not been evaluated in term of absolute efficiency measurements. This fact makes difficult the comparison between different methods published by different investigation groups.

When we claim that the efficiency is competitive, it is based on our experience working with different approaches inside out group.

In future works, comparison between different methods will be addressed.

Reviewer 2 Report

The submitted manuscript presents how to control aerial robotic swarm in surveillance missions, which employs a behavior-based algorithm. Finally, the method is experimentally tested in a laboratory environment. In the reviewer’s opinion there are still some concerns or questions which should be addressed by the authors before its acceptance:
(1)In this paper, a novel behavior-based method is proposed. The benefits of the method have been illustrated clearly. Are there other ways that the assumption of the condition can be further reduced?

(2) Information fusion provides a powerful tool to deal with uncertainty and external disturbance. For example, Human-Manipulator Interface based on Multisensory Process via Kalman Filters, A Markerless Human-Robot Interface Using Particle Filter and Kalman Filter for Dual Robots, Markerless Human-Manipulator Interface Using Leap Motion with Interval Kalman Filter and Improved Particle Filter. Brief discussions are helpful. The reviewer doubts if the authors can combine the sliding mode estimation technique to improve the robustness of the developed method.

(3) In conclusion part, more future works and challenges are recommended.

Author Response

We would like first to thank the reviewer for the comments provided. The comments have been here collected and answered one by one. Major changes have been highlighted. A link to a uploaded video has been added in page 20.

The submitted manuscript presents how to control aerial robotic swarm in surveillance missions, which employs a behavior-based algorithm. Finally, the method is experimentally tested in a laboratory environment. In the reviewer’s opinion there are still some concerns or questions which should be addressed by the authors before its acceptance:
(1)In this paper, a novel behavior-based method is proposed. The benefits of the method have been illustrated clearly. Are there other ways that the assumption of the condition can be further reduced?

The three main assumptions or hypothesis considered in this work are:

Position and velocity in global axis known Communications between agents Model of the quadcopter

Positioning errors have been considered, injecting a gaussian noise to the measured position (see Section 5.2).

Lose of messages have been also considered in Section 5.1.

Finally, the model considered has a variability in the parameters (see Table 1). For each simulation, the values of the model parameters has been withdrawn from a normal distribution considering that standard deviation.

In case of the shape of the surveillance area, or the possibility of changing the flying altitude, will be addressed in future works.

Is there any other hypothesis we could relax, or study to relax?

(2) Information fusion provides a powerful tool to deal with uncertainty and external disturbance. For example, Human-Manipulator Interface based on Multisensory Process via Kalman Filters, A Markerless Human-Robot Interface Using Particle Filter and Kalman Filter for Dual Robots, Markerless Human-Manipulator Interface Using Leap Motion with Interval Kalman Filter and Improved Particle Filter. Brief discussions are helpful. The reviewer doubts if the authors can combine the sliding mode estimation technique to improve the robustness of the developed method.

Position estimation will be improved in future works to overcome positioning errors and make the algorithm more robust. Anyway, in this work one of the aims was showing robustness against errors.

(3) In conclusion part, more future works and challenges are recommended.

The future works has been rewritten in one single paragraph. Changing the shape of the area and the flying altitude have been added.

Reviewer 3 Report

Abstract: 

"Among others properties...." --> Change the word "others" to "other"

Introduction:

Page 3, Line 101/102: Why the research work was limited to rectangular area? Does your algorithm suitable for flexible size and shape (rectangular or circular)? 

Page 4, Line 124:

4 2. The surveillance task --> Section title is not appropriate; In addition, I would recommend past tense to write this section.

Page 23: Figure 13: try to reduce the literature of the Figure caption.

8. Discussion:

Please change the section title to "Conclusion." Write the "Conclusion" section in a single paragraph. Don't use bullets or numbering in this section. Both qualitative and quantitative analysis should be summarized in this section. I would recommend past tense to write this section.

Author Response

We would like first to thank the reviewer for the comments provided. The comments have been here collected and answered one by one. Major changes have been highlighted. A link to a uploaded video has been added in page 20.

Abstract:

"Among others properties...." --> Change the word "others" to "other"

Changed. Thanks.

Introduction:

Page 3, Line 101/102: Why the research work was limited to rectangular area? Does your algorithm suitable for flexible size and shape (rectangular or circular)?

In previous works, the scenario has been characterized by the tuple (As/Na, Na, Vn, Rsfp, fA): area per agent, number of agents, nominal velocity, radius of sensor footprint, and shape factor of the area (lx / ly). Therefore, rectangular areas with different dimensions and shapes have been already studied in the past.

Depending on this tuple, the algorithm can be configured in order to achieve high efficiencies (see [17] for further details)

Changing the shape of the area would make more difficult this configuration, because the tuple that defines the scenario would be different.

Anyway the algorithm can work with area shapes without difficulties. It is just a matter of how to characterize the area.

In future works this will be addressed.

Page 4, Line 124:

4 2. The surveillance task --> Section title is not appropriate; In addition, I would recommend past tense to write this section.

Section title has been changed. We hope the new one is more appropriate.

The authors believe that the tense used (present and future) represents properly the work carried out.

Page 23: Figure 13: try to reduce the literature of the Figure caption.

The description has been reduced.

8. Discussion:

Please change the section title to "Conclusion." Write the "Conclusion" section in a single paragraph. Don't use bullets or numbering in this section. Both qualitative and quantitative analysis should be summarized in this section. I would recommend past tense to write this section.

The title has been changed to "conclusions and future works"

Bullets to enumerate characteristics of the algorithm, demonstrated with simulations and experiments, seem appropriate to the authors.

Future works have been rewritten without bullets.

A paragraph with a quantitative summary of the findings has been included.

When describing the work done, past tense is already used. For instance:

In this work, a behavior-based algorithm to carry out a surveillance task has been presented

Experiments have been also carried out...

When describing generically characteristics of the algorithm or general reflections obtained analyzing the results, we believe present tense is more appropriate, for instance:

...the algorithm allows the rest of the group to keep on with the surveillance.

...the algorithm presents two main drawbacks...

Round 2

Reviewer 2 Report

The submitted manuscript presents a control method for an aerial robotic swarm, which employs a behavior-based control method. Finally, the method is experimentally tested in a laboratory environment. After revision, some issues have been addressed and the paper has been significantly improved. In the reviewer’s opinion there are still some concerns or questions which should be addressed by the authors before its acceptance:
(1)The experimental setup should be described in more detail.

(2) The tasks of the robot should be made clear.

Author Response

We would like first to thank the reviewer for the comments provided. The comments have been here collected and answered one by one. Major changes have been highlighted.

(1)The experimental setup should be described in more detail.

Figure 11 has been added, in which it is shown a general arrangement of the elements (cameras, computers, etc) needed in the experiments. The paragraph where it is explained the arena+experiment hardware has been improved (page 20)

(2) The tasks of the robot should be made clear.

The task of the ground robots is just moving with linear trajectories throughout the area. As explained in page 23, second bullet of the list, each robot generates a goal point, and moves to it. When it reaches it, it generates a new one. The ground robots avoid the collisions between them.

Round 3

Reviewer 2 Report

After revision, some issues have been addressed and the paper has been significantly improved. In the reviewer’s opinion there are still some concerns or questions which should be addressed by the authors before its acceptance:
(1) The layout of the paper can be improved.
(2) The design of the experiment can be described in more detail.

Author Response

(1) The layout of the paper can be improved.

With this little information, the authors have been unable to know what the reviewer means with the "layout of the paper". Please, provide further details or examples.

(2) The design of the experiment can be described in more detail.

What does the reviewer mean with "the design of the experiment"?

Is it about the conditions of tests carried out? Is it about how exactly (hardware, software) the tests are carried out?

A sentence has been added on page 20.